# Vascularized Tissue Organoids

**DOI:** 10.3390/bioengineering10020124

**Published:** 2023-01-17

**Authors:** Hannah A. Strobel, Sarah M. Moss, James B. Hoying

**Affiliations:** Advanced Solutions Life Sciences, Manchester, NH 03101, USA

**Keywords:** organoid, vascularization, spheroid, stem cell, microvessel, vessel, endothelial, vascular network, perfusion, extracellular matrix

## Abstract

Tissue organoids hold enormous potential as tools for a variety of applications, including disease modeling and drug screening. To effectively mimic the native tissue environment, it is critical to integrate a microvasculature with the parenchyma and stroma. In addition to providing a means to physiologically perfuse the organoids, the microvasculature also contributes to the cellular dynamics of the tissue model via the cells of the perivascular niche, thereby further modulating tissue function. In this review, we discuss current and developing strategies for vascularizing organoids, consider tissue-specific vascularization approaches, discuss the importance of perfusion, and provide perspectives on the state of the field.

## 1. Introduction

Tissue organoids, or “mini organs”, are being fabricated with increasing physiological complexity. This makes them relevant tissue mimics that can provide insight into native cellular and tissue behavior in the context of both health and disease. Their small size makes them economical, and in many cases, limits regions of necrosis that are often present in thicker tissue models. Importantly, organoids can be fabricated from human cells. This provides a tremendous advantage over in vivo animal models, which are often not representative of cellular and disease behavior in humans. In addition, the 3D format of organoids creates a tissue environment that begins to better approximate the in vivo tissue space than 2D cell cultures [1,2,3,4]. In 3D, regulatory spatial arrangements of cells and more complex cell–matrix contacts are formed, which play integral roles in cellular communication and tissue function. Many different cell types, including osteocytes, smooth muscle cells (SMCs), endothelial cells (ECs), and tumor cells, for example, lose phenotypic characteristics in 2D culture as compared to 3D [5,6,7]. Thus, 2D cultures cannot mimic native tissue behavior and function with the same relevance that 3D organoids can. Another key advantage of organoids is the potential to be free from synthetic materials, containing only cells and their secreted matrix. While synthetic polymers have played many essential roles in other tissue engineering applications [8,9], they can negatively affect cellular behavior [10].

Because of these advantages, organoids are increasingly used on a commercial level in a variety of applications, including drug screening and disease modeling, across a wide range of tissue types. Currently, approximately 672 organoid-related patents exist, with 76% being specifically focused on organoid models [11]. The vast majority of these patents have been filed since 2014, with an exponential growth curve since then as the field progresses [11].

Despite these advantages of tissue organoids over other in vitro systems and animal models, many organoid strategies lack a functional microvasculature. The microvasculature is present in nearly every tissue in the body and serves multiple essential purposes. Firstly, it carries oxygen and nutrients, via the blood, throughout tissues to ensure the even distribution of the oxygen and nutrients, and tissue function. Secondly, the microvasculature is a distribution system for circulating cells, endocrine factors, and drugs. Thirdly, fluid flow through vessels and into the interstitial space provides key cues that modulate tissue function and health. Finally, blood vessels are a depot for a variety of different perivascular cell types in addition to endothelial cells (ECs) and smooth muscle cells (SMCs), including mesenchymal stem cells (MSCs), tissue-resident macrophages, pericytes, immune cells, and other stromal cells. These cells communicate with other cells in the tissue and play critical roles in modulating tissue homeostasis, phenotype, and dynamics. 

For these reasons, fabricating a mature and functional vasculature has been a large focus of the field. Towards this end, many recent advances in the field have involved increasing the vascular cellular complexity of organoids, resulting in improved tissue function overall, in addition to more native-like neovasculatures. This has been achieved through a variety of strategies, including advances in stem cell technology, more comprehensive cellular isolates, such as the stromal vascular fraction (SVF), or even fragments of whole microvessels, which are similar to SVF but retain their vessel structure in addition to cellular complexity. Finally, while the importance of perfusion and fluid forces has long been known, microfluidic technologies have recently enabled some degree of intraluminal flow within organoids. While this has been a large step forward, fully and consistently perfused in vitro constructs have yet to be achieved. In this review, we will discuss strategies for vascularizing tissue organoids in the context of the requirements for a vasculature to be both mature and functional, and discuss strategies for perfusing these vasculatures.

## 2. Organoid Fabrication Strategies

The term “organoid” refers to an aggregation of different cell types that communicate and function in a manner similarly to human tissue. This differs from “spheroid”, which is more generically used to describe any aggregate of cells. While the two terms are often used interchangeably in the literature, an organoid is more complex. This increased complexity facilitates the development of a more dynamic tissue-like environment involving complex signaling networks between different types of cells. For example, culturing stem cells in an aggregate form, in the presence of the correct signaling factors, can produce organoid models that contain epithelial cell-lined structures analogous to the gut [12], kidney glomeruli [13], or canaliculi of the liver [14]. 

There are a handful of methods used to build organoids, most of which rely on variations of a self-assembly approach [15,16,17]. An early developed approach is the “hanging drop” method, common in embryoid body culture, whereby drops of cell suspension are placed in a well plate which is then inverted (Figure 1A). Within each drop, cells will settle and aggregate into organoids. While simple, this method does require precise liquid handling, and there is a moderate risk of losing organoids during suspension dispensing and organoid formation. In perhaps the most common approach, cells are seeded in wells of a non-adherent U- or V-bottom well plate (Figure 1B). Gravity brings the cells together in the bottom of the plate, and over a few hours they will form numerous cell–cell contacts to form an aggregate. Over subsequent hours and days, cells will produce an extracellular matrix, further anchoring cells together, and reorganize into a more cohesive structure. The centrifugation of well plates can aid this process by forcing cells together, promoting aggregation. While easy to use, organoid sizes cannot be easily modified when they are formed this way. Custom molds and microplates can be used to form different-sized wells to control organoid size, and they often increase throughput compared to a traditional well plate format. Some cell types will also form aggregates when cultured in a rotating bioreactor, which encourages cellular interaction and aggregation. However, this provides less control over aggregate size and uniformity (Figure 1C) [15,16,17]. 

## 3. Vascularization Strategies

Native in vivo vascularization occurs through two general processes. The first, “vasculogenesis”, involves the formation of new blood vessels from single cells. This largely occurs during embryonic development, as new tissues and microvasculatures are forming at the same time. It is also the predominant method used in many tissue fabrication approaches, including organoid formation. Multiple cell types are combined into a tissue and over time form primitive endothelial tubes supported by a perivascular cell type. 

The second vascularization process is “angiogenesis”, or the growth of new blood vessels from existing blood vessels. During sprouting angiogenesis, new vessels arise from existing vessels either by sprouting (sprouting angiogenesis) or splitting (intussusception) to increase tissue vascularity. In adults, angiogenesis is the predominant vascularization process. Angiogenesis is highly responsive to the tissue environment, and is influenced by a variety of directional cues, such as oxygen gradients, and cellular, biochemical, and mechanical signals. Neovessels may grow towards and inosculate with other vessels, form branching structures, and ultimately form a neovascular network. Over time this network will become perfused, remodel, and mature into a functional microcirculation [18]. Vascularization strategies, which may utilize either vasculogenesis or angiogenesis, are summarized in Figure 2. 

### 3.1. Vascularization with Endothelial Cells

The most common method for vascularizing 3D tissues is to incorporate ECs, typically human umbilical vein endothelial cells (HUVECs), into the organoid during cell seeding. A vasculogenesis-like process then takes place to self-assemble the cells into a primitive microvasculature. This method has been used in perhaps hundreds of studies, and applied to nearly every organoid tissue type, including brain [19], liver [20], dental pulp [21,22], bone [23], cardiac tissue [24,25], and many others. When incorporated into 3D tissues, ECs will typically form a series of tube-like structures. However, these EC tubes are often unstable, and do not truly form a complete microvasculature due to the absence of the other multiple cell types comprising the microvessel wall. These additional cells are essential for mimicking microvessel phenotypic complexity (Figure 3). 

Still, it is important to note that while EC-only systems do not result in a complete microvasculature, their presence can improve tissue model function. For example, Garzoni et al. observed improved cardiomyocyte contraction when the organoids contained HUVECs, compared to cardiomyocytes alone [26]. Hiramoto et al. fabricated fibroblast spheroids for drug screening and saw improved drug sensitivity when HUVECs were incorporated [27]. Dissanayaka et al. added HUVECs to dental pulp organoids to increase the matrix production of dental pulp cells [22]. Examples like this are plentiful, where even just one additional cell type can improve overall tissue function. From a vascularization perspective, while the inclusion of ECs may not result in a functional network in vitro, they can accelerate vascularization after implantation [19,21,28]. Host cells that infiltrate an implanted organoid can associate with existing EC tubes and build viable vessels more quickly, helping to establish blood flow and reduce necrosis. 

### 3.2. Incorporation of Other Vascular Cells

To increase microvessel complexity, other cell types such as mesenchymal stem cells (MSCs), pericytes, or fibroblasts are included along with ECs in the organoid seeding steps [30,31,32,33]. These stromal cells can help stabilize EC tubes and contribute to the stromal compartment of the organoid. MSCs are perhaps the most commonly incorporated secondary vascular cell, as they play key roles in angiogenesis and vascular development. MSCs in their undifferentiated state secrete a spectrum of growth factors, are well established to promote angiogenesis, and often act as pericytes, stabilizing a microvasculature [14,34,35,36]. These qualities make them an effective addition to improve organoid complexity. 

Takebe et al. combined both MSCs and HUVECs with iPSC-derived hepatocytes to form liver organoids [14,37]. Over 4 days, the HUVECs organized into vascular-like structures with associated MSCs. Then, the organoids were implanted into a mouse model, where they inosculated with host vessels within 48 h and matured into a complex native-like microvasculature. Shah et al. added MSCs to EC spheroids intended for studying angiogenesis [38]. The added complexity afforded by the MSCs improved the physiological relevance of the organoids and resulted in different responses to angiogenic molecules than ECs alone. MSC-EC organoids are also under investigation for their potential regenerative applications. Hsu et al. recently performed an intracranial implantation of MSC-EC organoids into mice that had suffered an ischemic brain injury [30]. Interestingly, the addition of MSC-EC organoids resulted in reduced scar formation, enhanced motor control, neurogenesis, angiogenesis, and a smaller infarct volume, compared to controls without implanted organoids. These results are extremely promising, especially given that so few treatments are available clinically for ischemic strokes and other ischemic injuries [30]. These examples show the many different applications MSCs can be used for. However, while MSCs add an additional cell type and level of complexity to the in vitro microvasculature, they still do not make up for a lack of other vascular cell types compared to the native tissue environment. Thus, it is important to explore the inclusion of additional cell types as well. 

Another key vascular cell is the macrophage. Similar to MSCs, macrophages also stimulate angiogenesis and neovascular network formation, and may guide growing neovessels and facilitate inosculation [39,40,41]. However, macrophages have been used much less frequently than MSCs in organoid-based tissue modeling. This is largely because macrophages are much more challenging to isolate and culture, primarily due to their phenotypic plasticity and tendency to differentiate rapidly. Macrophages have a broad spectrum of phenotypes, and their pro-vascular effects are not uniform across this entire spectrum. Still, a small number of studies have incorporated macrophages into larger engineered tissues to either enhance vascularization or study macrophage–neovessel interactions [41,42,43,44]. While we were unable to find any studies that used macrophages to specifically enhance organoid vascularization, a handful have used macrophages in organoids to better understand tumor and immune biology in the context of macrophage interactions [45,46,47]. One such study was performed by Bingle et al., who implanted macrophage-containing breast cancer spheroids into mice [45]. While they did not examine vascular cells specifically in the organoid, they did observe an increase in angiogenesis in the surrounding tissues, compared to tissue in mice that had cancer-only organoids that did not contain macrophages. While challenges with in vitro culture have limited their use, the macrophage is another key player in tissue vascularization, and its inclusion in organoids in the future may further improve function and vascularization potential. 

### 3.3. Stem Cell-Based Vascularization Approaches

Stem cell technologies have made tremendous progress in recent years, making these approaches much more commonplace throughout the field of tissue engineering. Stem cells can be expanded further than primary cells, making them attractive as large numbers of cells are needed to build tissues or for high-throughput screening applications. After expansion, they can be differentiated into a variety of cell types, and often retain more functionality than primary cells, which tend to lose their phenotype when cultured [5,7,13,25]. 

There are three primary stem cell types, MSCs, induced pluripotent stem cells (iPSCs), and embryonic stem cells (ESCs). MSCs are found throughout the body but are commonly isolated from bone marrow or adipose tissue. They can differentiate into several different cell types, including bone [48], adipose [49,50], cartilage [51,52], smooth muscle [53], and others. In the context of organoid fabrication, MSCs may be used to derive certain cell types, or, more commonly, they can be used as a vascular cell, as discussed above. They are also being investigated in therapeutic applications [35,54,55]. However, MSCs are limited as to which types of tissue they can differentiate into. Because MSCs are most often added as a support cell similarly to pericytes, in this section, we will focus on organoids derived from iPSCs and ESCs.

ESCs are of embryonic origin and are pluripotent, capable of differentiating into any cell type. iPSCs begin as primary cells, often peripheral blood cells or fibroblasts, which are re-programmed (de-differentiated) into multipotent stem cells. At this point, they can then be differentiated into a wide range of cell types from all three germ layers [56]. iPSCs also present the tremendous advantage of enabling the development of patient-specific models [57,58]. Cells such as fibroblasts can be easily isolated from a patient sample, then re-programmed and differentiated into the desired cell type. While this strategy is far from widespread in clinical use, the idea that one could test the effects of different therapeutics on a specific person’s cells in a more complex model is highly attractive and could help mitigate current problems with patient-to-patient variability in response to different disease treatments. 

When deriving vascularized organoids from stem cells, there are two main approaches that can be taken to develop the needed complexity. One approach is to differentiate and purify multiple different cell types independently, and then combine them together to form a multi-cellular organoid. The second approach is called “co-differentiation”, where multiple cell types are differentiated concurrently. Here, it is expected that a differentiation protocol will result in the majority of cells becoming a certain cell type, say, hepatocytes, while a smaller percentage will naturally differentiate into ECs, fibroblasts, smooth muscle cells, or other supporting vascular and parenchymal cell types. While technically less precise, this method results in a more native-like spectrum of cell types and has proven to be effective for fabricating functional organoids with vascular structures.

For example, Homan et al. developed kidney organoids using an iPSC co-differentiation approach [13]. The organoids were composed of iPSCs that had been cultured in a microfluidic chip device, where they differentiated into both kidney and vascular cells. The chip device was used to expose organoids to flow. Further, Homan et al. compared a co-differentiation approach to one where primary ECs and fibroblasts were mixed in with differentiated iPSC-derived kidney cells. They found that the incorporation of primary vascular cells did not benefit vascular network formation compared to the co-differentiation approach, and in fact inhibited nephron formation within the organoids [13]. Takasato et al. used a similar co-differentiation approach to fabricate kidney organoids [59]. While they did not incorporate the use of flow, organoids still formed distinct structural features including collecting ducts, nephrons, and tubules. A vascular network was present with lumenized endothelial tubes surrounded by pericytes. While not a truly mature kidney, the structure did strongly resemble that found in fetal kidneys, making it a potential tool for studying kidney development [59]. 

Wimmer et al. fabricated vascular organoids from iPSCs with the goal of creating a vascular mimic [32]. While the microvasculature is often thought of as a part of a specific tissue, its cellular, structural, and functional complexity truly make it its own organ. Here, they aimed to use vascular mimic organoids to study the effects of diabetes on the microvasculature. The majority of the work presented was performed with a co-differentiation approach. iPSCs were differentiated following an EC differentiation protocol, but enough cells spontaneously differentiated into pericytes and other vascular cells to stabilize the networks of EC tubes. Similar to Homan et al., the authors also tested an individual differentiation approach, where ECs and pericytes were differentiated separately, purified, and combined into vascular organoids. The individual differentiation approach resulted in much less stable vascular networks that lacked a basement membrane or adequate pericyte-EC interactions. Networks formed through co-differentiation did have a basement membrane and plentiful direct pericyte–EC contacts. The success of co-differentiation approaches is likely due to the resulting greater spectrum of cells compared to engineering an organoid with only two to three cell types. This cellular complexity is critical for vascular function and stability. 

Overall, the use of stem cells to fabricate vascularized organoids is one of the more promising approaches towards creating effective in vitro models. Compared to other methods with primary cells, they have a higher capacity for proliferation and regeneration, they tend to be more functional than primary cells, and they allow for the production of patient-specific models. However, some limitations remain. As mentioned above, some cell types may require very different differentiation protocols, making them difficult to co-culture. Additionally, some differentiation protocols are variable and do not produce consistent results, or are inefficient to the point that they are impractical. Further refinements and advancements in stem cell differentiation protocols may prove to be highly beneficial towards the goal of making reproducible vascularized organoids. 

### 3.4. Staging Culture Conditions

A major challenge of in vitro cell culture is how to culture different cell types together that have different culture condition requirements. For example, vascular endothelial growth factor (VEGF) is often required to support endothelial growth and angiogenesis, but it may prevent MSCs from differentiating into certain cell types, such as adipocytes, thus complicating the task of developing vascularized adipose organoids [60]. These contradictions are extraordinarily common, complicating the fabrication of many multicellular constructs. Here, staging different types of culture medium at different times throughout culture can in some cases allow different cell types to differentiate. Or, vascular cells may be added at a later time into the culture, to avoid co-culturing entirely during certain differentiation stages. 

Staged culture conditions can be particularly important when cell types are of different germ origins. Cells can be derived from one of three different germ layers of an embryo—the mesoderm, endoderm, and ectoderm. Cells from the same germ layer are typically easier to co-culture in the same construct than those from different germ layers. The microvasculature is derived from the mesoderm, meaning that it tends to be more straightforward to co-culture vascular cells with other mesoderm-derived tissue types such as bone, cartilage, and muscle. Ectoderm-derived tissues such as the cortical tissue in the brain may be challenging to co-differentiate alongside vascular cells, and alternative methods may be needed. Ham et al. fabricated vascularized cortical organoids from ESCs using a co-differentiation approach with staged medium changes, making them the first group to co-differentiate an organoid with components of two different germ origins [61]. They did this by following a neural differentiation protocol, but added a high concentration of VEGF (50 ng/mL) for days 1–6 of culture, and then a lower dose of VEGF (25 ng/mL) after that [61]. VEGF is well established to support endothelial growth and differentiation, and also plays a key role in a developing brain [62,63]. They observed that ECs assembled into vessel-like tubes that expressed markers characteristic of the blood–brain barrier. Over time, however, vascular density decreased, which the authors speculate was due to a lack of perfusion [61].

Pham et al. utilized a coating approach with iPSC-derived cortical and ECs [64]. Cortical differentiation requires a 3D format, so iPSCs were aggregated into organoids, and then differentiated into cortical cells. After 34 days of differentiation, organoids were coated with Matrigel and iPSC-derived ECs, which had been differentiated separately. Over a 3-week culture period, the ECs had spread and began to infiltrate the organoid, forming vascular-like structures. After implantation, organoids that had been pre-vascularized rapidly inosculated with the host microvasculature and formed a microvascular network within the organoid. Constructs that had not been pre-vascularized did not form vascular networks and died. This shows the importance of pre-vascularization, even when constructs are to be implanted and vascularized in vivo. By adding ECs at a later time point, the authors ensured the EC differentiation process would not be negatively affected by the cortical differentiation process [64]. 

In a different approach, Sun et al. differentiated ESC-derived “vascular precursor” organoids separately from ESC-derived brain organoids [65]. After a certain amount of maturation and differentiation, they fused the two types of organoids together, which then formed a vascularized brain organoid. After fusion, the authors observed increased neural progenitor cells and reduced apoptosis compared to non-vascularized brain organoids. In addition, microglia formed in vascularized organoids that were activated with immune stimuli [65]. Lee et al. used a staged culture approach to fabricate kidney organoids. Pluripotent stem cells were differentiated for 16 days in 2D culture prior to organoid formation. Then, cells were seeded into microfluidic devices, where they aggregated into organoids. A slow fluid flow rate was applied, resulting in approximately 1 dyne/cm^2^ of shear stress application to the organoid surface, and possibly some degree of interstitial flow. VEGF was then added to half of the devices, to encourage some cells to differentiate into ECs and vascularize the organoid. They found that constructs cultured with the staged VEGF and flow protocol contained PECAM-positive ECs that assembled into vascular-like structures, although they were limited and only growing outside of the organoid, rather than infiltrating the interior. Overall, flow improved organoid maturation, differentiation and function, and resulted in organoids that were more responsive to certain drugs [66]. Thus, even though the kidney and ECs may be challenging to co-differentiate, the use of staged culture mediums combined with fluid flow resulted in a functional kidney model. Such strategies can be highly beneficial for fabricating organoids composed of cells with incompatible culture conditions. 

### 3.5. Isolated Microvessel Fragments

While incorporating individual cell types, such as ECs, fibroblasts, MSCs, or pericytes, has proven useful, they do not fully recapitulate the “microtissue” that is the microvessel. As discussed, the native microvessel contains a variety of cell and matrix types comprising the vessel wall proper and the perivascular niche. These cell types are in constant communication, modulating each other’s behavior and function, both within the microvessel and between the microvessel and the tissue cells. Additionally, this cellular complexity is essential in forming a perfused microvascular network, as it enables network adaptation and remodeling in response to cues such as hemodynamic inputs, metabolic tissue needs, and intervascular communication [18]. Importantly, these adaptive abilities promote the formation of networks with architectures uniquely suited for the tissue space, while meeting fundamental tissue perfusion requirements (Figure 4) [18]. 

Towards this end, Hoying et al. developed a strategy for isolating whole, intact microvessel fragments from discarded lipoaspirates [67]. Through the selective removal of adipocytes and stroma, the remaining microvasculature is fragmented to produce pieces of intact microvessels that retain their cellular complexity, native structure, and phenotypic plasticity [68,69]. When embedded in a collagen matrix, the microvessel fragments spontaneously sprout to form growing neovessels that interconnect to form a neovascular network [67,70]. When implanted, this network rapidly inosculates with the host circulation to produce a stable microcirculation [71]. Microvessel constructs have been used for a host of applications, including studying stromal cell and vascular precursor dynamics [72,73], angiogenesis–tissue biomechanics [74,75], imaging modalities to assess neovascular behavior [76], post-angiogenesis microvascular maturation and patterning [77], building tissue models [50,78], and pre-vascularizing implants [79]. We routinely use this MV system as an in vitro experimental assay platform to evaluate angiogenic factors [80,81], identify putative angiogenic agents [82], evaluate microvascular instability [83], determine MMP-related angiogenic activity [80], and define matrix dynamics during angiogenesis [74,76]. 

Recently, Strobel et al. applied this approach to vascularizing tissue organoids [50]. First, the isolated microvessel fragments were combined with MSCs in an organoid format. The microvessels generated a neovascular network throughout the tissue, which, when embedded in collagen, invaded and vascularized the surrounding matrix. Next, a staged differentiation approach was utilized to fabricate vascularized adipose organoids, as the adipocyte induction medium inhibited angiogenesis and the angiogenic medium did not support adipocyte differentiation. MSCs were differentiated into pre-adipocytes and then combined with human microvessel fragments to form organoids, which was followed by a transition to maintenance medium. The maintenance medium supported the continued differentiation of pre-adipocytes into mature adipocytes, as well as promoting vascular network formation (Figure 5). Interestingly, adipocyte organoids with microvessels had more insulin receptors measured by PCR and IHC when compared to adipocytes that were not cultured with microvessels. This indicates that the presence of the microvasculature, even in the absence of perfusion, may play a functional role, although further experiments are needed [50]. 

In addition to whole microvessel incorporation, others have tried to replicate the cellular complexity of the native vasculature using SVF, rather than one or two individual cell types. SVF is essentially all of the cells contained within the vascular niche, but digested further into single cells, such that the matrix structure is gone but the cellular complexity of the vasculature remains. Muller et al. fabricated adipose organoids entirely from adipose-derived SVF, with the expectation that it would provide not only vascular cells, but adipose progenitor cells as well [84]. Using this method, cells developed into mature adipocytes and contained early vascular networks. Networks contained visible lumens, and stained positively both for CD31 and alpha smooth muscle actin, indicating the presence of both endothelial cells and stabilizing pericytes [84]. 

**Figure 4 bioengineering-10-00124-f004:**
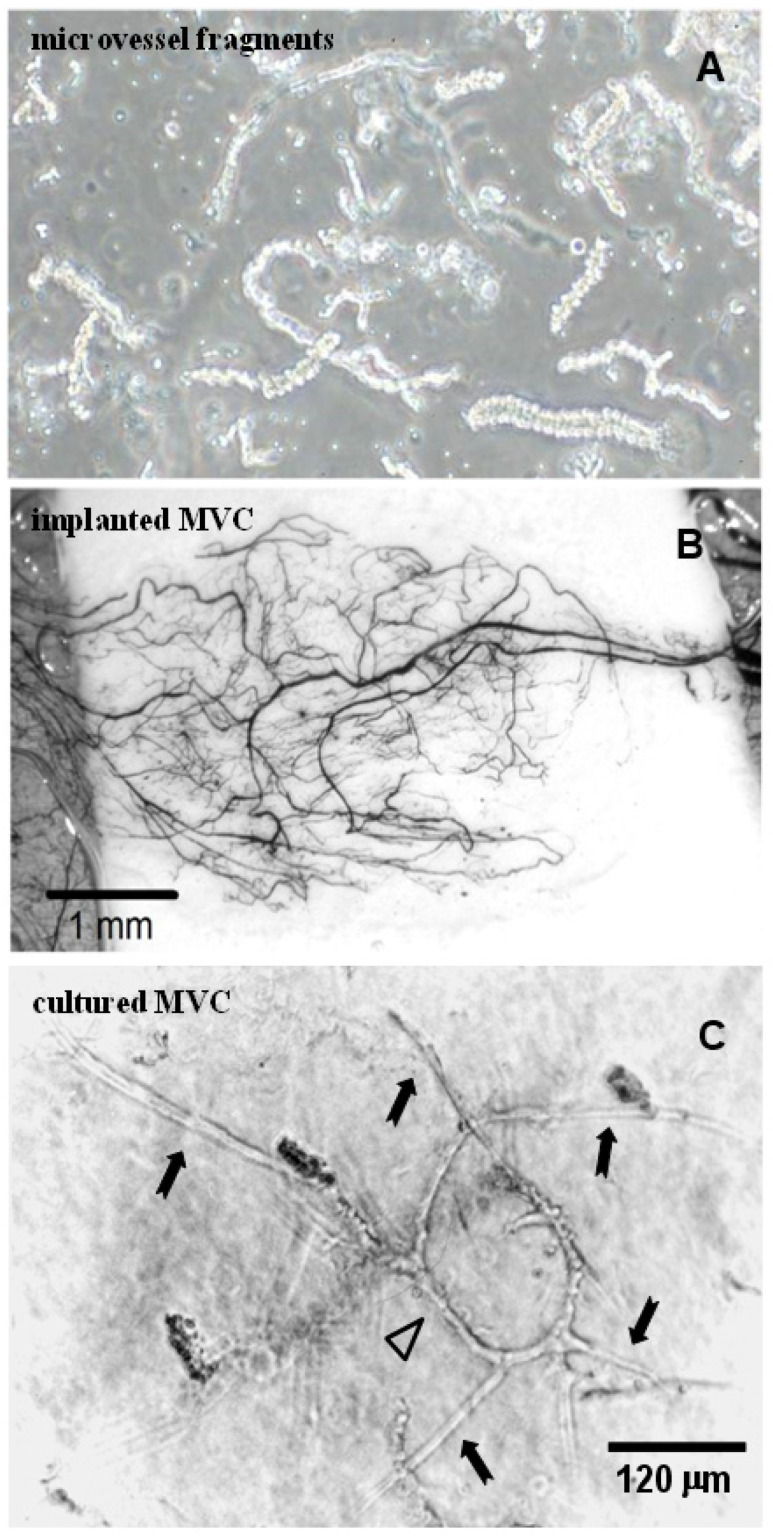
Microvessel Isolates. Individual microvessel fragments (**A**) are isolated from discarded lipoaspirates. Implanted vessels rapidly undergo angiogenesis to form a perfused network (**B**). When embedded in 3D matrices in vitro and cultured, sprouts (black arrow) form from parent vessel fragments (white arrow) (**C**), and grow to form a neovascular network. Figure adapted from LeBlanc et al. [85].

**Figure 5 bioengineering-10-00124-f005:**
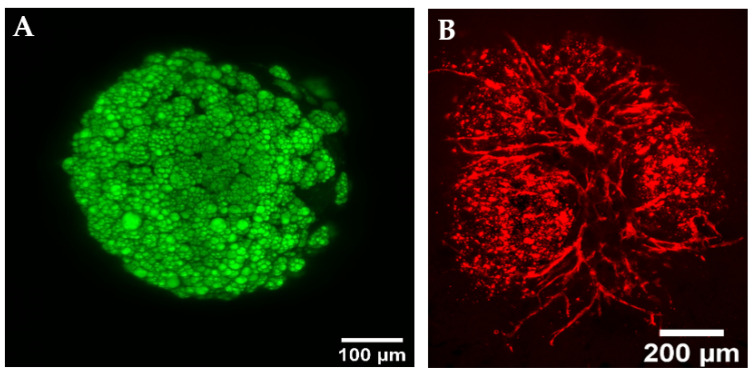
Vascularized adipose organoid. Microvessel fragments were combined with MSC-derived pre-adipocytes in an organoid format. 4,4-difluoro-4-bora-3a,4a-diaza-s-indacene (BODIPY) stain (**A**) shows lipid accumulation in differentiated adipocytes (**A**), while lectin stain shows the vasculature (**B**). Adapted from Strobel et al. [50].

While SVF contains the complete spectrum of vascular cells, it does not contain any matrix proteins, such as collagen or elastin, or the vascular structure. These matrix proteins can also affect cellular signaling. Nalbach et al. directly compared the vascularization potential of organoids with SVF to organoids with whole microvessel fragments [86]. Organoids with microvessels had more CD-31-positive and smooth muscle alpha actin-positive cells, and consequently more angiogenic sprouting, compared to those with SVF (with the same total number of cells). When implanted, organoids with microvessels also had higher rates of engraftment and vascularization [86]. This indicates that while cellular complexity is important, matrix composition and structure also influence vascularization. 

Takahashi et al. took this strategy a step further and incorporated whole tissue fragments into MSC-EC organoids [87]. Here, the tissue fragments contained intact microvessels, but also surrounding stromal and parenchymal environments. They applied this approach to a host of different tissue types, including brain, heart, kidney, liver, intestine, and others. Fragments of those specific tissues were aggregated into organoids with MSCs and ECs. The authors performed a thorough assessment of islet organoid function and compared tissue fragment organoids both with and without the added vascular cells. They found that the addition of MSCs and ECs contributed to islet function, enhancing organoid survival after implantation, insulin responsiveness, and glucose tolerance in mice. Tissue fragments alone were insufficient to enable the formation of vascular structures throughout the tissue, but they still brought in a spectrum of cell types that contributed to vascularization. MSCs and ECs alone cannot mimic the native vascular complexity, but the combination of tissue fragments and these vascular cells created a much more functional and vascularized islet organoid [87]. 

### 3.6. Matrix Considerations

While there is an emphasis on the critical aspects of cellular complexity in vascularization, it is important to note the role of matrix complexity in creating tissue models. As mentioned above, structurally intact microvessels have a higher vascularization capacity than the respective single-cell constituents [86]. It has been well-established that different matrix proteins can affect cellular phenotype and behavior [88,89,90], and can directly influence angiogenesis [91,92,93,94]. Towards this end, a handful of groups have experimented with adding matrix or other proteins to organoids to support vascularization. For example, Yarali et al. incorporated arginine–glycine–aspartic acid (RGD), a soluble integrin-binding peptide, into their HUVEC organoids [95]. The initial results indicate an upregulation of certain pro-angiogenic genes, including VEGF. This suggests their approach may accelerate angiogenesis if applied to tissue organoids; however, additional studies are needed to verify this [95]. 

Inamori et al. used a matrix coating approach to fabricate vascularized liver tissue [20]. They formed organoids from rat primary hepatocytes, and then coated them with collagen and HUVECs. While HUVECs invaded the spheroids regardless of the collagen coating, those with the additional collagen matrix formed a more uniform coating around the spheroid, which was important for their next step. Using an additive manufacturing approach, they stacked the spheroids together and fused them into one larger tissue. ECs between organoids formed primitive vascular networks throughout the large tissue. However, networks growing within individual organoids were limited [20]. This is one example where the addition of a collagen matrix helped modulate HUVEC behavior and vascularization. 

In Strobel et al., the impact of including stromal collagen in vascularized adipose organoids was assessed, in part, because isolated microvessels typically perform better in the presence of a fibrillar matrix [50]. Interestingly, including the type I collagen in the fabrication did not change the resulting vascular density of MSC-microvessel organoids. However, the vessels of adipose-microvessel organoids with collagen displayed a more mature vascular morphology than those without collagen, with more complex interconnected networks and branching structures [50]. It is possible that highly active cells such as MSCs secreted their own collagen so quickly that the exogenous matrix was unnecessary. Thus, it seems that the benefits of the additional matrix may vary depending on the tissue system. 

In some cases, adding a generic, off-the-shelf matrix protein may not be sufficient. In Kim et al., the authors fabricated kidney organoids using hPSCs and ECM from a decellularized kidney tissue [96]. They observed that over time, ECs in organoids with kidney ECM proliferated and formed interconnected tubular structures at a much greater extent than controls fabricated with Matrigel, which showed very little vascularization. When organoids were implanted into mouse kidneys, organoids with kidney ECM showed increased recruitment of host ECs, improved vascularization, and the better maturation of kidney glomerular-like structures compared to organoids without kidney ECM [96]. 

## 4. Tissue Specific Considerations

Every tissue in the body has its own unique structure, cellular composition, and function. Similarly, the microvascular phenotype is uniquely matched to each tissue. These differences manifest in the different vascular network architectures, EC phenotypes, and microvascular phenotypes. For example, reflecting the potential for exchange across the vessel wall, ECs in tissues such as muscle, adipose, or more extremely, the blood–brain barrier, can be “continuous”, where cells are held tightly together by numerous cell–cell junctions, limiting permeability. In endocrine tissues, or more extremely the liver, ECs can be “fenestrated” or “discontinuous”, with gaps between the cells that allow the exchange of larger molecules [97]. Establishing tissue- or organo-typic microvascular phenotypes in an organoid model captures more of the tissue biology, thereby modeling the native tissue environment with higher fidelity. 

### 4.1. Liver

Liver organoids are useful tools for modeling liver disease, and to screen the effect of pharmaceuticals on liver tissue and determine any potential liver toxicity. The liver is an endoderm-derived tissue responsible for removing harmful and potentially toxic substances in the blood, in addition to aiding with digestion and metabolism. It contains hepatocytes, the primary functional cells, as well as non-parenchymal cells (stromal cells) and a vasculature [98]. Liver sinusoidal endothelial cells (LSECs) are highly specialized (reviewed in [98]). They have a fenestrated phenotype and lack a continuous basement membrane, facilitating the exchange of blood plasma between the vasculature and the liver interstitial space. They have a much higher capacity to endocytose macromolecules and cellular components from the blood than other EC types, which aids in the liver’s filtration capabilities, and they contribute to the clearance of blood-borne pathogens and lipids, the immune response, and liver regeneration [98,99,100,101,102,103,104]. The dysfunction of LSECs, or more specifically the loss of the fenestrations and decreased permeability, can lead to hepatic fibrosis, liver disease, and cirrhosis [98,105,106,107,108]. 

Despite the importance of LSECs for healthy liver function, many liver models still utilize HUVECs as a vascular cell. This likely reflects the relative availability of HUVECs to isolate and purchase, compared to LSECs, which are much more technically challenging to work with. LSECs have been documented to lose their phenotype rapidly during culturing, although some studies indicate that 3D cultures or those with more precisely controlled microenvironments maintain their phenotypes for much longer [109,110,111]. This is not entirely surprising, as it is well-established that 2D models in general do not recapitulate the native 3D environment. 

A handful of recent studies have begun to incorporate LSECs into liver organoids instead of HUVECs. While many have shown good organoid function [112,113,114,115], none have thoroughly characterized LSEC morphology after longer culture periods, to prove that cells are in fact maintaining their fenestrated phenotype. This is widely considered a limitation of the field. However, one might argue that the functional outcomes with use of LSECs speak for themselves. Yap et al. compared organoids fabricated with either LSECs or HUVECs, and found that organoids with LSECs had much higher albumin secretion, improved organization of hepatobiliary structures, and improved organoid survival after implantation, compared to organoids fabricated with HUVECs [113]. While it is well established that vascular cells enhance organoid function, this demonstrates that the type and source of vascular cells also matters. While LSECs have historically been difficult to access, they are becoming increasingly available due to the advancement of genetic engineering technologies such as UpCyte and iPSC derivation protocols [112,116]. Hopefully, this will enable the expanded use of these cells in future liver models. 

### 4.2. Brain

The brain is an incredibly complex organ that presents an enormous challenge to model. Thus, a functional brain organoid has the potential to be an incredibly beneficial tool for disease modeling and drug screening. Most neurological disorders are very poorly understood, and there is a strong need to gain a better understanding of brain physiology and to develop new therapeutics to treat neurological pathologies. Tremendous progress has been made in recent years towards fabricating organoids from different regions of the brain. As with many tissue systems, methods that have included vascular cells have all shown improvements in brain function, particularly when organoids are perfused [64,117,118,119]. Perfusion strategies largely involved implantation, although Wang et al. utilized a chip device that applied interstitial flow [120]. As with all tissue types, future strategies should focus on combining both vascular network formation techniques and in vitro perfusion methods to achieve a truly functional organoid that is both vascularized and perfused, with both interstitial and intraluminal flow.

A key consideration with brain tissue, as with many tissues, is endothelial phenotype. Similar to kidney and liver ECs, brain ECs have their own unique phenotype. However, unlike the liver and kidney, brain ECs completely lack fenestrations and instead have many tight cell–cell junctions and adhesions, giving them a very low vessel wall permeability and resulting in the “blood–brain barrier” (BBB). The BBB functions to tightly control and regulate molecule movement between the blood and brain interstitial spaces within the central nervous system through a spectrum of specific transporters [121]. 

While brain ECs have been isolated, it can be challenging to culture them and maintain the BBB phenotype. However, there are examples in which the derivation of a BBB-like microvasculature has been formed by non-brain ECs. For example, Shi et al. seeded organoids containing iPSCs and HUVECs to fabricate vascularized cortical organoids [19]. Over an 83-day period, the authors observed that HUVECs began to adopt a BBB-like morphology as the cortical cells matured. This suggests that the cortical organoid environment provided sufficient BBB-related cues to the ECs in the organoid [19]. Similarly, Nunes et al. combined adipose-derived isolated microvessel fragments with precursors of astrocytes in a tissue construct. The construct was then implanted, where it matured into a functional microcirculation [72]. As compared to constructs prepared with fat-derived microvessels alone, those mixed with the astrocyte precursors developed into a microvasculature with tighter permeabilities and an upregulated expression of GLUT-1, a BBB EC marker [72]. Thus, in the absence of tissue-specific vascular cell sources, it may be possible in some cases to induce tissue-specific phenotypes in organoid microvasculatures, provided the necessary cues are present within the tissue environment. 

### 4.3. Kidney

The kidney is a complex organ that carries out important homeostatic functions, most notably the filtration of toxins from the blood and the maintenance of fluid homeostasis. Reflecting this, kidneys are extremely well vascularized and receive roughly 20% of cardiac output. Because the microvasculature is such a large and important component of the kidney, any vascular dysfunction can be enormously consequential for overall kidney function [122]. Structurally, kidneys contain thousands of specialized functional units called nephrons. Each nephron consists of a glomerulus, which is essentially the filtration unit, and a series of tubules, which collect waste and empty into a collecting duct that exits the nephron. When blood flows into the kidney, it first goes to the glomerular capillary bed for filtration. ECs comprising these vessels are very flat and highly fenestrated, allowing water and certain waste products to be filtered out, while retaining cells and proteins [122,123]. Glomerular capillaries contain several unique subtypes of ECs; some contain large fenestrations similar to those in the liver, some contain small fenestrations uniquely seen in the kidney, and others contain fenestrations with a diaphragm. Each of these types of ECs contribute to glomerular filtration, although exactly how is not fully understood [122,123,124]. After leaving the glomerulus, blood flows to the peritubular capillary bed, which associates with the different tubules in the nephron. While ECs are morphologically different in peritubular capillaries from those in glomerular capillaries, they are also highly fenestrated and contain diaphragms. These capillaries are responsible for delivering oxygen and nutrients to the surrounding tissue, and may reabsorb some of the contents filtered out in the glomerulus [122,123]. 

Despite the structural complexity of the kidney, the field of organoid fabrication has made significant progress towards creating relevant organoid models. A number of different groups have used pluripotent stem cells with or without fluid flow to fabricate kidney organoids that contained distinct compartments resembling different structures, such as the glomerulus, tubules, and vasculature, although the EC phenotype has not yet been examined [13,59,125,126]. A complete model combining the glomerulus and the collecting system has not been reported. It is important to note that these complex structures and functional compartments are not the result of advanced engineering approaches with pre-patterned tubules or microvasculatures, but by mimicking the biological environment as much as possible through the use of derived multi-cellular populations and interstitial flow. The literature has shown that “form follows function”, and if the right biological signals are provided, development will progress on its own, as it does in the body, into a functionally and structurally mature organoid. 

### 4.4. Muscle

The three types of muscle, skeletal, smooth, and cardiac muscle, are different tissues that perform contracting functions for different purposes. Their microvasculatures, while similar in many ways, are also uniquely adapted to suit each tissue’s needs. The microvasculature in skeletal muscle is highly organized, with each arteriole supplying about 15–20 capillaries. These capillaries form dense networks that are arranged parallel to each individual muscle fiber [127,128]. This high density is necessary as the metabolic needs of skeletal muscle can change rapidly and become very high during muscle work [129]. These rapid changes in oxygen demand require the microvasculature to be highly dynamic and specialized [130]. Capillary networks can change and adapt their morphology and density over time to suit the changing needs of the tissue, as it shifts between fast and slow twitch fibers, and as overall muscle mass increases. In fact, skeletal muscle adaptation is one of the few scenarios where angiogenesis occurs in a non-pathological setting in adult humans [131]. 

From an organoid perspective, modeling angiogenesis within skeletal muscle organoids may be beneficial to the end of understanding skeletal muscle adaptability and associated pathologies. However, the focus of the field has largely been on fabricating larger constructs for volumetric muscle loss therapies (reviewed in [132]), and progress towards building organoids specifically has been limited. This is perhaps largely due to the fact that skeletal muscle requires mechanical stretching to mature properly [133,134], and this is difficult to achieve in a traditional organoid format. Still, there is great opportunity for the development of engineered skeletal muscle containing microvasculatures that better recapitulate the mature and adaptable state seen in in vivo.

Similar to skeletal muscle, cardiac muscle also has an extremely dense microvascular network. Because the heart is responsible for pumping blood throughout the body, it has a continuously high metabolic need. Vessels also must be adaptable, like skeletal muscle, as exercise can dramatically increase the amount of blood flow needed by the tissue. This need is met both by increases in vascular density, through angiogenesis, and by changes in vascular morphology, such as increases in vessel diameter [135,136]. In contrast, a decrease in capillary density is often associated with disorders such as ischemic or dilated cardiomyopathy [137]. The field of cardiac organoids has grown considerably, and a handful of groups have used them specifically for modeling cardiomyopathy-related diseases [138,139,140]. For example, Filippo et al. fabricated organoids from iPSC-derived cardiomyocytes, cardiac ECs, and cardiac FBs. Cardiomyocytes were either healthy or contained an MYH7 mutation to induce cardiomyopathy [139]. Cardiomyopathic organoids were found to have an irregular beating pattern compared to healthy control organoids, as is typical in a clinical pathological setting [139]. Similarly, Archer et al. developed an iPSC-derived triculture organoid model [140]. They subjected organoids to numerous cardiotoxins and observed clear morphological changes with treatment and loss of structural proteins. This indicates the potential utility of the model for detecting the cardiotoxic effects of experimental pharmaceuticals [140]. While these studies do include a vascular component, more progress is needed towards building a complete, mature microvasculature. Given the strong association between angiogenesis, capillary density, and cardiac function, it will be essential moving forward to emphasize the development of a dynamic microvasculature to recapitulate native disease progression more accurately. 

Smooth muscle contracts around a variety of non-cardiac hollow organs, including the stomach, bladder, airways, and blood vessels. Each of these organs serves a different purpose, and consequently the microvasculature may have unique adaptations. For example, vessels in the bladder need to be able to withstand repeated distension. This is achieved through a very dense region of vessels at the apex of the bladder that forms loops and folds to accommodate stretching [141]. A number of groups have fabricated bladder organoids, most commonly for the purpose of modeling bladder cancer [142]. However, it is challenging to accurately model bladder microvasculature in an organoid format, given that a primary feature is its ability to stretch and distend. Still, there is room for increased emphasis on the vascular component, as many recent models do not include ECs [142,143]. 

Every organ contains a microvasculature that is uniquely suited to meet the needs of the relevant tissue being modeled. Thus, as the field of organoid fabrication continues to progress, we must continue to strive not just to vascularize organoids, but to produce vasculatures that match tissue specific functions. 

## 5. Mechanics of the Microvasculature

### 5.1. Perfusion and Fluid Mechanics

There are two primary types of fluid flow through a tissue, interstitial and intravascular (or intraluminal) flow. Interstitial flow is fluid movement outside of the blood vessels through the tissue space or interstitium. The source of interstitial fluid is the blood plasma, as fluid moves from the blood into the interstitium through the capillary walls (the degree of which is a function of capillary wall permeability). Fluid is returned to the blood either back through the capillary walls or through the lymphatic vasculature. Interstitial flow rates are low (0.1–2 µm/min [144,145]), which produces low-level shear forces that influence local cellular behavior and establish convective gradients, which modulates angiogenesis [146,147,148,149]. 

Intravascular flow is the flow of blood and plasma within the vasculature. Flow and shear rates vary widely depending on the location within the vascular tree, but are invariably larger than interstitial values. These shear forces play a powerful role in maintaining and modulating the health and function of ECs and other vascular cells [146,147]. For example, shear forces stimulate the production of nitric oxide (NO) through the increased expression of NO synthase [150]. NO decreases the expression of thrombogenic markers, which are indicators of EC damage and can cause thrombosis [151,152]. It also regulates endothelial permeability, differentiation, and cytokine production [153,154,155,156,157]. NO modulates angiogenesis, which is essential for organoid vascularization [158,159], and finally, it affects the phenotype and behavior of supporting vascular cells, such as MSCs, SMCs, and pericytes [153,154,155]. Pericytes and SMCs wrap circumferentially around blood vessels, outside of the EC layer. These cells contract and relax to regulate intraluminal pressure and flow. NO mediates this behavior; increased NO causes relaxation and consequentially increases vessel diameter and lowers pressure, while decreased NO does the opposite and increases pressure [160]. Changes in pressure can, independently of shear and NO, modulate both EC function and tissue angiogenesis [161,162,163]. Thus, the inclusion of fluid flow, and the associated mechanical forces, in organoid fabrication and applications can be an important step in using organoids to model native tissues. 

### 5.2. Perfusion Strategies

In the context of organoids, perfusion is relevant in multiple ways. Fluid flow can be applied to the exterior of the organoid, producing low levels of shear around the outside of the organoid, and, potentially, to the organoid interstitium depending on the organoid density and microenvironment. Modeling native tissue perfusion, however, requires intravascular flow through a properly constructed microvasculature within the organoid. This is much more challenging, as it requires fluid flow through both inlet and outlet avenues within the organoid and a mature vascular network. To have a perfused vascular network within an in vitro organoid, with intravascular flow, is perhaps the “holy grail” of in vitro organoid work. 

There are two primary methods that have been used to establish perfusion within organoids. The simplest method is to implant the organoid and allow the body to remodel and mature any pre-formed vascular structures, inosculate host and implant vessels, and begin flowing blood through the implant. This strategy of letting the body drive the biology and establish blood flow is by far the most effective, and has been widely used to create functional, vascularized, truly perfused organoids [19,28,30,37,119]. It is especially beneficial for long-term experiments, as organoids without adequate perfusion may exhibit poor survival over longer periods compared to implanted and perfused organoids [119]. It should be noted however, that while this method is often extremely effective when applied to small tissue organoids, vascularization may occur too slowly to be broadly applied to larger engineered tissues. Even among organoids, the presence of pre-existing vascular structures may be necessary, depending on the organoid type [19,28]. 

While implantation is a highly effective method to both vascularize and perfuse tissues, one of the key advantages of organoids is the ability to perform analyses in vitro, without the cost and ethical concerns of using animals. As a result, many groups have been working to establish in vitro perfusion, largely through the use of microfluidic devices or chips. These constructs are commonly referred to as “organ-on-a-chip” technologies. Here, tissues are cultured in a specially designed chamber on the microfluidic chip, and then fluid is pumped through the chamber using a series of channels and inlet and outlet ports. This approach has been extensively used to study the effect of fluid flow on 2D cultures. However, 2D cultures are not necessarily as informative of the in vivo environment as 3D cultures. Next, “vascular mimics” evolved, which are 3D tubular structures formed by ECs, with or without a stromal cell, in a low-density matrix [164,165,166]. These studies have provided another level of information, but still do not recapitulate the cell- and matrix-dense environment of an organoid, or of a native tissue. 

Organ-on-a-chip technologies may or may not contain organoids in the traditional sense, meaning spheroidal 3D tissues; however, regardless of format, they are still “mini organs”. These technologies have been used for drug screening, disease modeling, and understanding tissue biology, within a wide range of tissues. The key advantage of chip technologies is the ability to introduce fluid forces, and precisely control fluid flow on the micro scale. As discussed above, fluid forces play an essential role in modulating tissue function, and even small changes in these forces can affect cell signaling. In addition, microfluidic organ-on-a-chip devices in some cases can be arrayed together to create body-on-a-chip systems. This can inform us how different organ systems may influence each other in the context of different diseases or medications. Many thorough reviews exist on organ-on-a-chip devices [167,168,169,170,171]. Thus, here we will focus on efforts to achieve a mature, native-like microvasculature (as opposed to EC-lined microfluidic channels) with intraluminal perfusion.

At this time, many groups have succeeded in delivering interstitial flow through organoids [66,148,149]. While efforts to establish intraluminal perfusion in the organoid via a microvasculature have achieved some degree of perfusion, a fully perfused tissue has not been achieved in vitro [13,117,172]. Homan et al. and Cakir *et al*., for example, were able to demonstrate some intraluminal accumulation of fluorescent beads delivered to the organoids [13,117]. However, organoids were not uniformly perfused throughout; the organoids contained regions that did not have any perfusion, and the process was not consistently repeatable [13,117]. Regardless of intraluminal flow, Homan et al. did see a substantial increase in vascularization in the presence of flow, compared to static cultures (Figure 6) [13]. Nashimoto et al. used a microfluidic device with feeder vessels that grew into and vascularized tumor organoids [172]. Perfusion of the larger feeder vessels resulted in fluid flow through the vessel within the organoid and large functional improvements. However, only one or two vessels had actually grown into the organoid, and they did not form a network [172]. All three of these studies, despite their limitations, present enormous steps forward towards the goal of in vitro perfusion. This is, perhaps, the greatest challenge in the field of 3D tissue modeling. Having a fully functional, fully perfused microvasculature will be the largest step towards having a truly biomimetic in vitro tissue model. 

### 5.3. Other Mechanical Forces

While fluid forces are often the most discussed form of mechanical stimulation with regards to the microvasculature, many other forces can modulate vascularization and vascular function. External loading forces such as stretching or compression alter cellular behavior and signaling throughout the tissue microenvironment, which can in turn regulate angiogenesis. For example, compression forces have been shown to promote angiogenesis in bone [173,174]. Within each tissue, smaller mechanical forces and mechanical signals can be created by the cells and matrix themselves, which modulate angiogenesis. Each growing neovessel consists of a “tip” cell and one or more “stalk” cells. The tip cell is at the front of the neovessel, guiding the vessel through the matrix. As the tip cell moves, it deforms and reorganizes the surrounding matrix, both by degrading it through the use of matrix metalloproteases (MMPs) and by exerting pulling forces on the fibrils as it moves [175]. Many factors affect the speed at which a tip cell can travel through the matrix, and in what direction. A higher matrix density, for example, will slow down vessel growth and tissue vascularization [176]. Independently of density, matrix stiffness can also affect EC function and angiogenesis, with stiffer surfaces tending to have a pro-angiogenic affect [177]. Mechanical drivers of angiogenesis are further reviewed in [178]. While these factors—cell derived forces, matrix mechanical properties, and external loading—are not typically considered during organoid fabrication, it is important to be aware that they can also play a role in neovascular growth. 

## 6. Conclusions and Future Perspectives

The field of organoid fabrication has made tremendous progress in recent years. Still, gaps remain in our ability to fully model native tissues with organoids. A key challenge is how to increase organoid tissue complexity, particularly via vascularization, without over-complicating organoid fabrication and use. Approaches such as stem cell co-differentiation, whole microvessel incorporation, and the incorporation of whole tissue-specific cell isolates and matrix additives are steps towards achieving this goal. Future studies should continue to focus on mimicking the biological complexity seen in vivo, remembering the interplay between form and function. The cell–cell and cell–matrix signaling that occurs within tissues between parenchyma, vasculature, and stroma plays essential roles in both vasculogenesis and angiogenesis. Once biological complexity is achieved, providing necessary mechanical cues in the form of intravascular perfusion will also be essential. While in vitro perfusion is a target that remains elusive, impressive progress towards this goal in recent years suggests that we may be close to solving this problem. Future advances in microfluidics, in combination with biological factors, will move us towards this goal. However, it is still necessary to have a mature, stable neovascular network for perfusion to be achieved and maintained. To have a fully vascularized, in vitro perfused tissue organoid of any tissue type will be an enormously powerful tool, with potential to greatly improve our understanding of tissue biology in health and disease, thereby accelerating the development of therapeutic interventions.

## Figures and Tables

**Figure 1 bioengineering-10-00124-f001:**
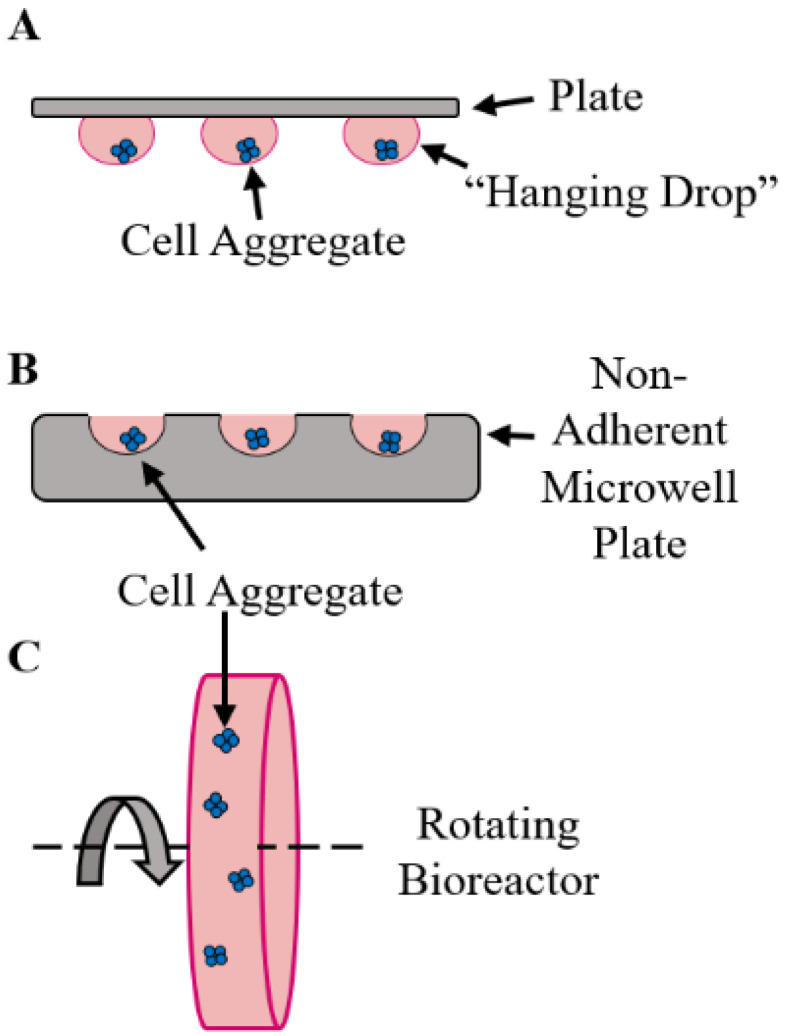
Common methods of organoid fabrication. The “hanging drop” method involves placing drops of cell suspension on a plate and inverting it such that cells settle and aggregate within the drop (**A**). In recent years, the use of non-adherent microwell plates has become perhaps most frequent (**B**). Plates are commercially available, but can be custom made as well. In some cases, the use of a rotating bioreactor can also promote cell aggregation (**C**).

**Figure 2 bioengineering-10-00124-f002:**
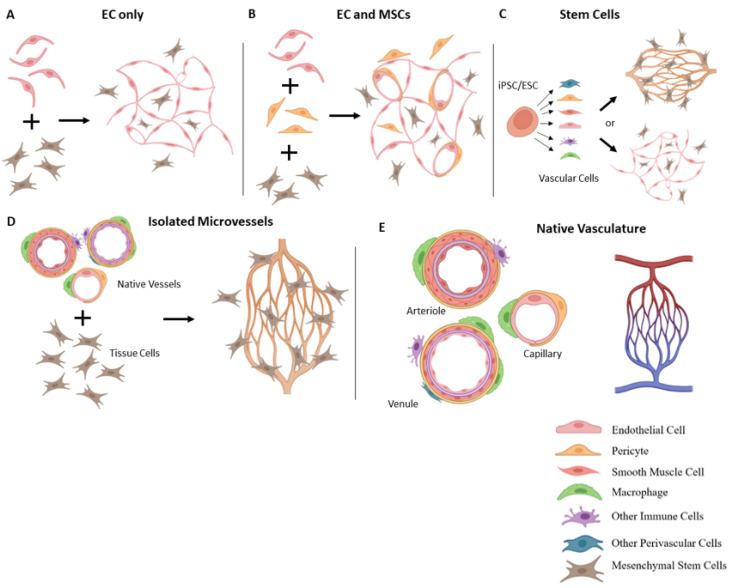
Vascularization Strategies: EC-only approaches frequently result in primitive EC tubes, rather than mature vasculatures (**A**). Adding in perivascular cells, such as MSCs, can add some stability to the network (**B**). Some groups have more recently derived tissues from stem cell aggregates (**C**). Stem cells can, given the correct stimuli, differentiate into a spectrum of cell types, which can mimic the cellular complexity found in vivo. This cellular complexity, when it is achieved, can lead to a more mature neovascular phenotype. Alternatively, incorporating whole microvessel fragments can also bring in the required cellular and structural complexity (**D**) needed to mimic the native vasculature (**E**).

**Figure 3 bioengineering-10-00124-f003:**
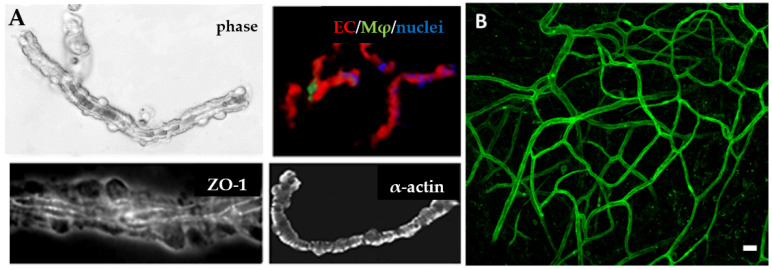
Native microvasculature. Microvascular fragments are shown in (**A**) with phase contrast and fluorescent microscopy. Multiple cell types, including ECs, macrophages, and pericytes can be seen on vessel fragments. Native microvasculatures contain a complex, hierarchal structure, shown in (**B**). Panel B adapted from Strobel et al. [29].

**Figure 6 bioengineering-10-00124-f006:**
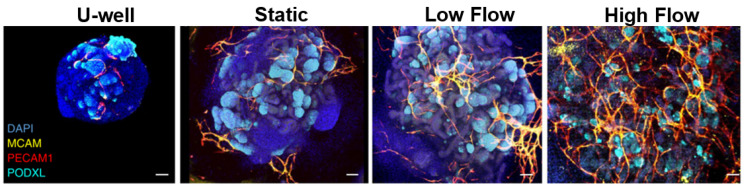
Microfluidic organoid culture. Microfluidic devices in Homan et al. applied fluid flow to organoids, which increased vessel growth. While flow was largely external, some degree of intraluminal perfusion was evident. Adapted from Homan et al. [13].

## Data Availability

No new data were created or analyzed in this study. Data sharing is not applicable to this article.

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
