# Peer review of "Vascularized Tissue Organoids"

_bioengineering, 2023, doi:10.3390/bioengineering10020124_

Round 1

Reviewer 1 Report

A well-written review by the authors. These are the comments about the manuscript:

1. Add a diagram that shows the important factors for the vascularization of tissue organoids in the Introduction section. This will help readers understand the overall importance and relation between the manuscript's content.

2. Add a figure for the organoid fabrication strategies to help the readers to visualize the methods.

2. The label for Figure 1 (A) is missing from the image. Replace the image with a new one because the text on the image is blurry and difficult to read.

3. Line 172. Use the abbreviation for iPSC and ESC since the full terminology has been written on line 163.

4. Figure 2. The word "in vitro" in the figure legend should be italicised.

5. Move Figure 4 to before the section "Other Mechanical Forces".

Author Response

Reviewer 1

  1. Add a diagram that shows the important factors for the vascularization of tissue organoids in the Introduction section. This will help readers understand the overall importance and relation between the manuscript's content.

Due to the usual absence of intravascular fluid flow in most organoid applications, the most influential factor in vascularizing tissue organoids is mimicking the cellular complexity found in vivo. We have added a schematic (Figure 2) showing different strategies for vascularizing tissue organoids, which speaks to this point.

  1. Add a figure for the organoid fabrication strategies to help the readers to visualize the methods.

We have added a schematic to manuscript (Figure 1) showing different organoid fabrication strategies.

  1. The label for Figure 1 (A) is missing from the image. Replace the image with a new one because the text on the image is blurry and difficult to read.

This image has been replaced with a higher resolution image and improved labelling.

  1. Line 172. Use the abbreviation for iPSC and ESC since the full terminology has been written on line 163.

We have corrected this and used the abbreviations for iPSC and ESC throughout.

  1. Figure 2. The word "in vitro" in the figure legend should be italicised.

We have corrected this and checked that in vitro is italicized throughout.

  1. Move Figure 4 to before the section "Other Mechanical Forces".

Figure 4 is now located before this section. 

Reviewer 2 Report

Dear authors! Thanks for the review article.

Good work, where the main aspects of this area are covered.

Please follow the design of the MDPI journals.

Author Response

Dear authors! Thanks for the review article. Good work, where the main aspects of this area are covered. Please follow the design of the MDPI journals.

Thank you for your feedback. We have carefully checked that the manuscript follows all format requirements laid out on the MDPI author instructions website. 

Reviewer 3 Report

Comments                                                                                       Date: 25/12/2022

Manuscript Number: 2147119

The submitted review manuscript entitled “Vascularized Tissue Organoids” described a detailed description and compiled information on biomedical applications such as disease model and drug screening to control a disease. Authors have followed a right way review methodology, consistent description and gleaned informative data for readers working in the same filed. Authors conducted various literature survey to collected required data. However, there are many comments need to be addressed. Such as;

1.     In introduction section, I suggest to expand introduction section with recent advancements. These information must be clearly described with major findings on vascularized tissue organoid for biomedical applications followed by advantages and disadvantages.   

2.     Authors should present various responsive polymers used for the model and organ related model. These must be explained in table form. This would help reader to find various material for the purpose with their limitation and fate (biological metabolism). I recommend authors to compile details of various in vitro and in vivo model used clinically or reported in literature.   

3.     Authors could not mention methodology adopted for thorough review of literature from different sources. What strategies were adopted?.

4.     The sentence “While SVF does contain a full spectrum of vascular cells, it does not contain any of the matrix proteins or vascular structure, which can also affect cellular signaling” must be rewritten. This is unclear and confusing. The sentence “They applied this approach to a host of different tissue types, including brain, heart, kidney, liver, intestine, and others, taking fragments of that specific tissue and aggregating them into organoids with MSCs and endothelial cells” is too long.

5.     There are several reported patents and their major outcomes on organ models. I suggest to include a table of patent related information (bibliometric analysis of global organoid technology patents, referred to doi: 10.1016/j.isci.2022.104728).

6.     Authors missed to include information on organoids on-a-chip and their benefits. This must be included as a separate section. BODIPY must be expanded. There are several abbreviations used without defining them in the manuscript text body.  

7.     Organoids are important means for 3-D culture in vitro. How these overcome the shortcoming of 2D culture and restored the specific organ functionality?. This should be addressed in revised file.

8.     There are lots of grammatical and typo errors in the manuscript drafting. Many sentences are too long and difficult to follow. I found many incomplete sentences for rewriting. There are several typographical errors.    

9.     I suggested various corrections in the manuscript. Therefore, abstract and conclusion need to be rewritten. In the sentence “The field of cardiac organoids has grown considerably, and a handful of groups have used them specifically for modeling cardiomyopathy related diseases [134-136]” authors cited three different references. Please explain findings from each of them.    

10.   A new section for future perspective must be added in the revised version of manuscript.

11.  In figure 1, figure 1A is missing. The figure must be replaced with high resolution and clear image. Caption of figure 1 needs to be expanded to describe each image section (by proper labelling). Authors should keep consistency in italic form of “ in vitro” in the manuscript.

Author Response

Reviewer 3

  1. 1.In introduction section, I suggest to expand introduction section with recent advancements. These information must be clearly described with major findings on vascularized tissue organoid for biomedical applications followed by advantages and disadvantages.   

      We have added statements in the introduction summarizing recent advancements in the field. In addition, recent advancements are described throughout the manuscript. Major findings are discussed with each individual study throughout the manuscript.

  1. Authors should present various responsive polymers used for the model and organ related model. These must be explained in table form. This would help reader to find various material for the purpose with their limitation and fate (biological metabolism). I recommend authors to compile details of various in vitro and in vivo model used clinically or reported in literature.   

We agree that polymers have many essential applications in the field of tissue engineering. However, a key feature of most tissue organoids is the absence of synthetic polymers; they are fabricated entirely from cells and their secreted matrix. We have added a sentence to the introduction speaking to this point, as it is a very important distinction. New polymers are being developed, but at this point without widespread adoption within the field of organoid fabrication. We reference recent reviews where readers can go to learn more about polymers in other tissue engineering applications.

  1. Authors could not mention methodology adopted for thorough review of literature from different sources. What strategies were adopted?.

      A thorough review of the literature was conducted, with search terms including “vascularized organoid”, “vascularized spheroid”, “perfused organoid”, as well as many variations of those terms. Manuscripts were carefully read, and any papers relevant to the topic or organoid vascularization and perfusion were cited or discussed. The focus of the search was original manuscripts, although recent organoid related reviews were explored as well.

  1. The sentence “While SVF does contain a full spectrum of vascular cells, it does not contain any of the matrix proteins or vascular structure, which can also affect cellular signaling” must be rewritten. This is unclear and confusing. The sentence “They applied this approach to a host of different tissue types, including brain, heart, kidney, liver, intestine, and others, taking fragments of that specific tissue and aggregating them into organoids with MSCs and endothelial cells” is too long.

      We have rephrased these sentences to clarify and shorten them.

  1. There are several reported patents and their major outcomes on organ models. I suggest to include a table of patent related information (bibliometric analysis of global organoid technology patents, referred to doi: 10.1016/j.isci.2022.104728).

      We agree that it is beneficial to add a discussion on commercialization of organoid models. We have added a brief discussion of patents and commercialization to the introduction and referenced the mentioned manuscript. However, there are over 600 patents relevant to tissue organoids, and to properly tabulate relevant data, without excluding a significant amount of information would take up a significant portion of the manuscript. Given that the focus of the manuscript is on organoid vascularization, we believe including such a table would be outside of our scope.

  1. Authors missed to include information on organoids on-a-chip and their benefits. This must be included as a separate section. BODIPY must be expanded. There are several abbreviations used without defining them in the manuscript text body.  

      We agree that organ-on-a-chip is an important topic. This was a key component of our perfusion section, as a main advantage of these systems is the ability to provide fluid flow. We have expanded this section to further discuss the benefits and accomplishments of organ-on-a-chip technology. However, this is far too large of a subject to fully review within the present work. Thus, we have also added references to several recent reviews specific to organ-on-a-chip technologies to direct readers to additional reading.

We have checked that all abbreviations are defined.

  1. Organoids are important means for 3-D culture in vitro. How these overcome the shortcoming of 2D culture and restored the specific organ functionality?. This should be addressed in revised file.

This critical topic of 3D vs 2D has been addressed by others in a number of quality reviews. Our intent was to not recap this content, but to emphasize the nature and role of the vasculature in these 3D applications. Regardless, we have briefly expanded on this topic in the revised document in the introduction section.

  1. There are lots of grammatical and typo errors in the manuscript drafting. Many sentences are too long and difficult to follow. I found many incomplete sentences for rewriting. There are several typographical errors.

      We have checked and corrected any typographical errors throughout and shortened sentences where appropriate.    

  1. I suggested various corrections in the manuscript. Therefore, abstract and conclusion need to be rewritten. In the sentence “The field of cardiac organoids has grown considerably, and a handful of groups have used them specifically for modeling cardiomyopathy related diseases [134-136]” authors cited three different references. Please explain findings from each of them.    

      We have expanded on this section to discuss the findings of these studies. We have ensured that the abstract and conclusion accurately reflect the content of the manuscript.

  1. A new section for future perspective must be added in the revised version of manuscript.

      We have expanded the conclusion to highlight future perspectives. In addition, future perspectives are provided with each section. We have added and expanded on this throughout the manuscript.  

  1. In figure 1, figure 1A is missing. The figure must be replaced with high resolution and clear image. Caption of figure 1 needs to be expanded to describe each image section (by proper labelling). Authors should keep consistency in italic form of “ in vitro” in the manuscript.

This image has been replaced with a higher resolution image and improved labelling. We have checked that in vitro is italicized throughout.

Round 2

Reviewer 3 Report

Authors rectified all raised comments and revised the manuscript. I recommend the article for publication